# Mindfulness, Physical Activity and Sports Participation: Validity Evidence of the Lithuanian Translation of the State Mindfulness Scale for Physical Activity 2 in Physically Active Young Adults

**DOI:** 10.3390/bs13100820

**Published:** 2023-10-05

**Authors:** Migle Baceviciene, Rasa Jankauskiene, Vaiva Balciuniene

**Affiliations:** 1Department of Physical and Social Education, Lithuanian Sports University, 44221 Kaunas, Lithuania; vaiva.balciuniene@lsu.lt; 2Institute of Sport Science and Innovations, Lithuanian Sports University, 44221 Kaunas, Lithuania; rasa.jankauskiene@lsu.lt

**Keywords:** mindfulness, exercise, psychometric properties, monitor and acceptance theory (MAAT), motivation, body image

## Abstract

The aim of the present study was to examine the psychometric properties of the Lithuanian translation of State Mindfulness in Physical Activity (SMS-PA-2) in a sample of physically active students. A total sample of 539 students from universities and colleges (50.3% men; mean age 23.3 ± 7.2 years) were asked to provide demographical data, report physical activity (PA) and PA habits and fill in measures of trait awareness, autonomous motivation in physical activity, trait body appreciation, trait body functionality appreciation and disordered eating attitudes and behaviours and self-esteem via an online anonymous survey. Students also provided information about their participation in organized team sports, organized individual sports, recreational sports and home exercise. Results. The SMS-PA-2 replicated the original four-factor structure, with good internal consistency (except for the Accepting Mind subscale). Invariance analyses across sex groups revealed an acceptable fit of the configural, metric and scalar models. However, in the multi-group analysis, metric invariance and scalar invariance were not confirmed. The SMS-PA-2 was positively associated with leisure-time PA, PA habits, more self-determined motivation for exercise, trait awareness, trait body appreciation, trait body functionality appreciation and self-esteem. A negative association was observed between the SMS-PA-2 and disordered eating attitudes and behaviours. Home exercisers and students engaged in team sports demonstrated lower levels of state mindfulness in PA than those engaged in organized individual and recreational sports. The Lithuanian version of the SMS-PA-2 is a reliable and valid instrument for measuring state mindfulness in PA. This instrument is recommended for researchers who aim to investigate the role of state mindfulness in the PA in Lithuanian-language-speaking samples of young adults. Future studies should explore the Lithuanian version of the SMS-PA-2 by asking participants to fill questionnaires in immediately after the PA session, and measurement invariance between sexes should be further tested.

## 1. Introduction

### 1.1. The Conception of Mindfulness and Monitor and Acceptance Theory (MAT)

Mindfulness is conceptualized as a process of bringing a certain quality of attention and awareness to one’s experience (the experience a person undergoes) and doing it by focusing on the present moment, purposively and non-judgmentally (how a person experiences it) [1,2,3,4,5]. In other words, mindfulness entails becoming an observer of one’s own thoughts, feelings and bodily sensations without being judgmental [2]. The attention that is directed towards stimuli (i.e., body feelings) with judgment and criticism does not represent mindfulness [6]. A rapidly increasing number of publications show that mindfulness is positively associated with general well-being, life satisfaction, happiness, autonomous motivation, enjoyment, self-esteem, self-efficacy, sleep quality, nutritional behaviour and nature connectedness and negatively with stress, anxiety, depression, chronic pain and substance abuse [7,8,9,10,11,12,13,14]. 

Based on the conception of mindfulness, monitor and acceptance theory (MAT) [10] was developed. MAT proposes that mindfulness works through two components: attention monitoring and acceptance. These two components are addressed in the most scientific conceptualization of mindfulness and are the main skills that are developed in mindfulness training programmes. The first tenet of MAT is that attention-monitoring skills enhance awareness of the present-moment experience and might improve cognitive functioning and heighten affective experiences (both positive and negative). Acceptance skills modify the way one relates to present-moment experiences and regulate reactivity to affective experiences. In other words, together with monitoring, acceptance increases psychological well-being and reduces negative reactivity (i.e., anxiety, depression and stress) [10]. The results of a recent study suggested that monitoring alone marginally predicted depression, anxiety and stress, whereas acceptance strongly predicted both reductions in ill-being variables and increases in life satisfaction and happiness [13]. 

Mindfulness might be conceptualized as an individual difference (trait mindfulness) as well as a momentary experience (state mindfulness) that can vary among people in different contexts and times [15]. People high in trait mindfulness experience mindful states more frequently than people who are low in trait mindfulness. Thus, state mindfulness might be a modifiable target of interventions when people are trained to pay attention to the present moment in a particular way. Higher states of mindfulness may manifest over time as an increased trait of mindfulness. In their systematic review [15], proposed that mindfulness has a hierarchical three-level bidirectional structure. State mindfulness might be divided into situational and contextual mindfulness. Situational mindfulness represents the level of mindfulness that an individual experiences at a specific moment and varies as a function of the context and time. Contextual mindfulness represents an individual’s typical level of mindfulness within a specific context (i.e., physical activity). Global mindfulness (trait) represents a general disposition towards mindfulness across varied contexts and moments in daily life [15]. Scholars suggest that these three levels should be distinguished in physical activity research [15].

### 1.2. Mindfulness in Physical Activity

Typical organized sports activities are goal-oriented rather than process-oriented and might involve a relative disconnect between body and mind. Conversely, other activities that incorporate regulated breathing and focused attention (such as yoga, Pilates and tai chi) are considered as being focused towards mindfulness [16]. One study found that trait mindfulness was positively related to yoga but negatively to aerobic physical activity [17]. There is a lack of research assessing state and trait mindfulness in different physical activities [18]. Thus, one of the goals of the present study was to compare state mindfulness in physical activity between groups of different sports and exercise types. 

State mindfulness studies are not well represented in the physical activity literature. Most of the empirical studies in the physical activity domain focused on trait mindfulness and did not assess mindfulness as a state construct [15]. Exploring state mindfulness in physical activity is an important issue since state mindfulness might facilitate autonomous motivation through increased self-control, self-regulation and enhanced positive body image in physical activity [19,20,21,22]. Higher states of mindfulness in physical activity might help people to overcome problems with self-control, self-regulation and body image concerns and increase internal motivation, further enhancing trait mindfulness and the maintenance of physical activity. Thus, it is important to test state mindfulness in physical activity and to have reliable and valid instruments for measuring it. Therefore, in the present study, we will further discuss the psychological mechanisms through which mindfulness affects physical activity motivation and body-image-related issues and then present the rationale for the validation of the Lithuanian version of the State Mindfulness Scale in Physical Activity-2 (SMS-PA-2) [1].

### 1.3. The Psychological Mechanisms through Which State Mindfulness Affects Physical Activity Motivation

Research shows that trait or dispositional mindfulness is positively associated with physical activity through various psychological mechanisms [8,15,18]. One of the most important mechanisms is enhanced autonomous motivation for physical activity. According to self-determined motivation (SDT) [21], a positive relationship between more autonomous forms of motivation and exercise exists, and intrinsic motivation is predictive of long-term exercise adherence [23]. Specific pathways through which mindfulness might support autonomous physical activity motivation go through the satisfaction of basic psychological needs (BPNs). Individuals are naturally inclined to make choices that satisfy their innate needs (BPNs) to feel autonomous and competent. Non-judgmental and open awareness of physical body sensations might support feelings of competence and autonomy in physical activity (i.e., by selecting different physical activities or lowering the intensity of physical activity when feeling discomfort). This helps individuals to be more connected with themselves, feel more satisfaction with physical activity, and continue physical activity for long periods, gaining in autonomy and competence over time. Finally, mindfulness might help individuals to have more self-determined motivation when following physical activity plans that are externally prescribed by coaches. 

According to SDT, the more internal goals of exercise, such as exercising for enjoyment and satisfaction, are linked with prolonged physical activity [23]. Experiencing positive sensations and feelings of competence in exercise are internal rewards that might be noticed when the exerciser directs attention with awareness to internal emotions and physical sensations in an open and non-judgmental manner. Awareness of one’s internal experiences (such as thoughts, feelings and sensations) and external environmental conditions is a major variable that supports autonomous motivation [21]. Research shows that state mindfulness in physical activity is associated with autonomous physical activity motivation [6,24]. State mindfulness is related to more internal exercise goals (i.e., health and mood), but not to appearance-enhancing exercise goals [6]. 

Awareness of internal thoughts, feelings and sensations when making decisions is in line with one’s values, needs and interests. Mindful awareness could enhance the acceptance of negative and uncomfortable thoughts and negative sensations that might occur during physical activity, especially in exercisers with little experience, individuals with body image concerns and overweight people (i.e., shame, discomfort, pain, fatigue, exertion). It might help to improve self-control [20] and encourage people to sustain physical activity in the short and long term. People with elevated mindfulness might be more aware of their feelings and thoughts in a non-judgmental way. Thus, for example, facing stress and anxiety during physical activity might prevent them from dropping out of exercise since they can focus on coping and managing negative emotions more effectively [25]. Research showed that mindfulness training increases executive functioning, which controls and directs cognitive processes for working memory, planning, decision making, self-regulation and many other goal-directed behaviours [26]. These cognitive processes might strengthen individuals’ self-regulation and their ability to follow their physical activity goals by constantly having them in mind and effectively coping with dropping out from exercise [19].

### 1.4. Mindfulness, Body Image, Disordered Eating and Physical Activity

Mindfulness might be a mediator between physical activity and positive body image. A positive body image is associated with greater physical activity [22]. There is evidence that trait mindfulness (awareness and acceptance) is associated with a more positive body image (operating as body appreciation) [14] and negatively associated with state body surveillance [6]. Body surveillance is a self-objectification behaviour involving a persistent focus on one’s body and predicting how others will evaluate it [27]. It is the opposite of body functionality, which refers to respecting and honouring the body for what it is capable of doing [28]. Monitoring external and internal sensations in a non-judgmental manner might shift attentional focus from body appearance surveillance towards body functionality in physical activity. Higher levels of mind and body acceptance in exercise might be associated with greater body appreciation, which is associated with greater physical activity and intrinsic exercise motivation [22,29]. 

A recent meta-analysis suggested that mindfulness was inversely related to body dissatisfaction, binge eating and emotional eating. Non-judging and acting with awareness had the strongest negative relationships with eating disorder psychopathology [30]. Another recent meta-analysis concluded that yoga interventions demonstrated a small but significant effect on eating disorder psychopathology, a moderate-to-large effect on binge eating and bulimia and a small effect on body image concerns [31]. These findings suggest that higher levels of monitoring and awareness may lead individuals to lower behavioural automaticity, which might lead to lower disordered eating, while higher levels of acceptance and non-judging may promote acceptance of body image, thoughts and emotions about one’s body and food without trying to avoid or change them (30). Exploring the associations between state mindfulness, physical activity and body-image-related issues is quite a new research area, with most studies being conducted using Western European samples. Therefore, more research is needed, and one of the objectives of the present study is to provide more knowledge on this issue. 

### 1.5. State Mindfulness in Physical Activity Scale-2 (SMS-PA-2)

To assess state mindfulness, the State Mindfulness Scale (SMS) was developed as a self-report measure [32]. It was designed to measure individuals’ perceived level of attention to, and awareness of, their current experience during a specific period of time (i.e., in the last 15 min) and in a specific context (i.e., meditation or physical activity). The SMS had 21 items and 2 subscales, namely body and mind. The scale showed good psychometric properties in various populations [5,32] and was back-translated into, and validated in, a number of languages [5]. Nevertheless, this measure was not fully suitable to measure mindfulness in a specific context such as physical activity. 

To measure state mindfulness in physical activity, the State Mindfulness Scale-PA was developed [6]. This instrument consists of 12 items and a bi-factor model reflecting the mindfulness of the mind (6 items) and body (6 items) (Mindfulness of Mind and Mindfulness of Body) as specific factors, and the general mindfulness factor was supported. To date, the first version of the SMS-PA has been validated for use in Spanish, Turkish and Brazilian cultures [24,33,34,35]. Researchers have employed the SMS-PA with young people [36]. The SMS-PA measures focus on the attentional and awareness aspects of state mindfulness in physical activity. However, the aspects related to acceptance, non-judgment and openness were not well captured in this instrument. According to MAT [10], monitoring and acceptance are skills that interact and underlie the positive effects of mindfulness. Acceptance refers to accepting the present experience with openness and receptivity and without criticism. Recent evidence suggests that acceptance is even stronger than monitoring and is associated with indicators of well-being and appears to be the most important dimension of mindfulness [13]. Therefore, the second version of the instrument was developed. 

In the second version of the SMS-PA-2, acceptance items were included to better represent core elements of mindfulness and MAT [1]. Specifically, the SMS-PA 2 extends the original version of the scale by including items that test acceptance of the body and mind in physical activity. This scale is aligned with MAT and the mindfulness conception [3,10]. The instrument consists of 19 items. Four subscales were identified in the second version of the instrument: Monitoring of the Mind (six items), Monitoring of the Body (six items), Accepting the Mind (three items) and Accepting the Body (four items). A 15-item version also showed good psychometric properties. In the present study, we aimed to validate a Lithuanian translation of the 19-item version of the SMS-PA-2. To the best of our knowledge, this manuscript presents one of the first trials to test psychometric properties of this instrument in another language other than English. 

Measuring state mindfulness in physical activity is important since it represents a dynamic mental state that is related to time and activity [5]. Validation of this instrument would let us assess the effects or correlates of mindfulness in different contexts (i.e., different types of physical activity) and at distinct levels of competence (novice or experienced exercisers) and observe the changes in state mindfulness as the outcome of various interventions. Mindfulness-based interventions (MBIs) might be helpful for individuals not benefiting from exercise lifestyle interventions (e.g., overweight, obese people) [18] and individuals with body image concerns [30]. Having a reliable instrument measuring state mindfulness in physical activity is also important in measuring the effectiveness of interventions that aim to test the effect of mindfulness-based physical activity on positive body image and its correlates.

### 1.6. The Present Study

The aim of the present study was to examine the psychometric properties of the Lithuanian translation of the SMS-PA-2 in a sample of physically active student-aged women and men. We expected that the Lithuanian version of the SMS-PA-2 (SMS-PA-2-LT) would replicate the original four-subscale structure. In addition, we aimed to assess the measurement invariance between sexes, expecting to establish it. Next, we aimed to evaluate the convergent and discriminant validity of the instrument using measures of trait awareness, physical activity, motivation in physical activity, positive body image (operating as body appreciation), disordered eating attitudes and behaviours and self-esteem. Since there is a lack of research assessing mindfulness in various physical activities and sports [18], the final aim of the present study was to compare SMS-PA-2 scores in groups of different types of sport involvement (organized team sports, organized individual sports, recreational sports and home exercise). 

We expected that mindfulness would be positively related to physical activity level, a physical activity habits index, autonomous motivation, positive body image (operating as body appreciation), body functionality appreciation and self-esteem, and negatively to disordered eating attitudes and behaviours. In the present study, we also expected that mindfulness would be greater in participants of organizational and recreational sports than in those exercising at home. Previously, a higher level of mindfulness was observed in organized sport participants than in individuals not participating in any sports [24].

## 2. Materials and Methods

### 2.1. Participants

Altogether, *n* = 1114 students started the survey; 759 agreed to participate and completed it, while 12 students refused to participate. After applying the inclusion criteria described in the “Procedure” section, 539 questionnaires were confirmed for the final analysis in this study. Study participants were from seven Lithuanian universities (*n* = 486) and two colleges (*n* = 53); 271 (50.3%) were men and 268 (49.7%) were women, representing all study areas: technical, medical/health, social sciences and humanities. The majority of the university students were studying in first-cycle study programmes (78.3%). The mean age of the study participants was 23.3 ± 7.2 years (range 18–44 years).

### 2.2. Procedure 

All data were collected via an online survey between November 2022 and May 2023. The survey was implemented through the Survey Monkey platform. All questions were set as mandatory. This prevented us from missing data. This study was approved by the Social Research Ethics Board of Lithuanian Sports University (protocol number SMTEK-131, 30 October 2022). Participants were recruited using a non-probabilistic sampling method. Inclusion criteria were as follows: age ≥ 18 years, studies at any state or private Lithuanian university or college, Lithuanian language spoken, regular participation in any organized or recreational sports for not less than half a year and participation in an exercise session no longer than two weeks ago. Only those students who confirmed being able to recall their experience during the last exercise session were recruited to this study. In the present study, 20.4% of the sample had exercised on the same day, 29.8% on the previous day, 26.3% on the previous two to three days, 9.3% within one week and 14.2% exceeding one week. 

Prior to completing the survey, participants were introduced to the study aims and study measures and were told the approximate time needed to complete the survey (20–25 min). The survey form was restricted to accepting only one response from the same IP address. After providing digital informed consent, participants were directed to the measures described in the Materials and Methods section. Those who refused to participate or did not meet the inclusion criteria were acknowledged, and the survey was terminated for these individuals. Additionally, study participants could end the survey at any point by closing their browser, with their responses being excluded from further analysis. After obtaining permission from the administrative units of each participating university/college, the link to the anonymous survey was distributed to the study participants in their classrooms by a trained researcher.

The translation of the SMS-PA-2 into Lithuanian was first performed by two professional translators, and then it was back-translated into English by another two professional translators. The final translation was reviewed by the scientists working in the field of psychology and discussed with the translators to obtain a comparable meaning of content and to ensure item clarity and semantic equivalence. A pilot study with 11 exercisers from health and fitness centres was conducted, and some minimal language corrections were made based on the feedback of the pilot study participants. Guidelines and recommendations for the translation of the instrument were addressed in the process [37]. The translated version of the SMS-PA-2 is presented in Appendix A. 

### 2.3. Measures

The Lithuanian translation of the State Mindfulness Scale for Physical Activity-2 (SMS-PA-2-LT) [1] was used to assess state mindfulness in physical activity, specifically monitoring mind and body and acceptance of mind and body. The scale was described previously. The 19-item scale uses a five-point Likert scale with responses ranging from “0 = not at all” to “4 = very much”. The original instrument supported the four-factor structure, construct validity and internal consistency [1]. The internal consistency of the instrument was good. For the general scale, Cronbach’s α was 0.87; for the Monitoring Mind subscale, it was 0.83; for Monitoring Body, it was 0.86; for Accepting Mind, it was 0.56 (without item No. 15, Cronbach’s α was 0.63); and for Accepting Body, it was 0.76.

The Lithuanian translation of the Philadelphia Mindfulness Scale (PHLMS) [38] was used to assess trait awareness. This scale measures distinct facets of present-centred awareness (10 items) and acceptance (10 items). In the present study, only the awareness subscale was used. Examples of items of the awareness subscale include: “When talking with other people, I am aware of their facial and body expressions” and “When I walk outside, I am aware of smells or how the air feels against my face”. Items were rated on a five-point Likert scale ranging from 1 (never) to 5 (very often), and subscale scores were computed by totalling the response options. Higher scores represent higher levels of awareness. In the present study, the internal consistency of the scale was good: Cronbach’s α = 0.83.

The Lithuanian version of the Leisure-Time Exercise Questionnaire (LTEQ) [39] was used to assess physical activity. Participants of this study were provided with examples of light, moderate and strenuous physical activities and asked to report the frequency of each session lasting 15 min or longer during the last week. The frequency of light physical activity was multiplied by 3, moderate activity by 5 and strenuous activity by 9. The scores of each intensity level were summed, and the final physical activity score was obtained. A higher score represents higher physical activity. The Lithuanian version of the LTEQ had been used in previous studies with young adults [40]. 

The Lithuanian version of the Self-Report Habit Index (SRHI) [41] was used to assess the strength of physical activity habits. The SRHI might be used to assess the strength of habits in various behaviours, including nutritional habits, sedentary behaviour habits, etc. In the present study, the measure was adapted to test physical activity habits. The SRHI starts with the stem “Behaviour x is something…” followed by 12 items. An example item is “Physical activity is what I have been doing for a long time”. Items are followed by seven-point Likert response options that range from complete disagreement (1) to complete agreement (7). Items are summed and averaged to get an overall SRHI score that ranges between 1 and 7, with a higher score representing a higher strength of physical activity habits. The Lithuanian version of the scale demonstrated adequate psychometric properties and a unidimensional factor structure [42]. In the present study, the internal consistency of the scale was good: Cronbach’s α = 0.92.

The Lithuanian translation of the Behavioural Regulation in Exercise Questionnaire-2 (BREQ-2) [43,44] was used to assess five different levels of autonomy-related behavioural regulation in physical activity (amotivation, as well as external, introjected, identified and intrinsic motivation). The questionnaire consists of 19 items with response options on a five-item Likert scale that range from 1 (“not true for me”) up to 5 (“very true for me”). The Cronbach’s α values for this investigation were as follows: for the amotivation subscale, 0.81; for the external motivation subscale, 0.82; for the introjected regulation subscale, 0.74; for the identified regulation subscale, 0.66; and for the intrinsic regulation subscale, 0.87. The Lithuanian translation of the questionnaire demonstrated acceptable psychometric properties in previous studies [45,46]. 

The Lithuanian version of the Body Appreciation Scale-2 (BAS-2) [47] was used to assess positive body image in students. The unidimensional scale consists of 10 items on a five-point Likert scale, with possible answers ranging from “Never” (1) up to “Always” (5). Higher scores indicate greater body appreciation. Examples of the items include: “I feel good about my body” and “I appreciate the different and unique characteristics of my body”. The Lithuanian version of the scale demonstrated good psychometric properties [48]. The internal consistency of the scale in the present study was good (Cronbach’s α = 0.95).

The Lithuanian version of the Functionality Appreciation Scale (FAS) [49] was used to assess functionality appreciation, that is, appreciating, respecting and honouring the body for what it is capable of doing, extending beyond mere awareness of body functionality. The scale consists of seven items on a Likert-type scale ranging from “Strongly disagree” (1) up to “Strongly Agree” (5). A higher score reflects greater body functionality appreciation. Examples of the items on the scale include: “I appreciate my body for what it is capable of doing” and “I appreciate that my body allows me to communicate and interact with others”. The Lithuanian version of the scale was validated in previous studies [50]. The internal consistency for the present study was good (Cronbach’s α = 0.92).

The Lithuanian translation of the Rosenberg Self-Esteem Scale (RSES) [51] was used to assess the self-esteem of students. The RSES is the most widely used measure of self-esteem for adult populations. The scale is composed of 10 items, 5 of which are negatively worded. Items on a four-point Likert-type scale response format range from “Strongly Agree” (4) to “Strongly Disagree” (1). A higher score represents greater self-esteem. Item examples include: “I feel that I have a number of good qualities” and “I certainly feel useless at times”. The Lithuanian version of the RSES has been widely used in adult populations. In the present study, Cronbach’s α = 0.89.

The Lithuanian version of the Eating Disorder Examination Questionnaire-6 (EDE-Q-6) [52] was used to assess disordered eating attitudes and behaviours. The EDE-Q-6 consists of 28 items with response options ranging from “Never” (0) up to “Always” (6). To calculate the final score, items 1–12 and 19–28 are used. The questionnaire consists of several subscales; however, in the present study, we used only the final score. A higher score indicates that the respondent expresses greater disordered eating attitudes and behaviours. The psychometric characteristics of the Lithuanian translation of the questionnaire were acceptable [53]. In the present study, the internal consistency of the questionnaire was good (Cronbach’s α = 0.93).

Participation in sports was assessed with two single-item indicators. First, students were asked if they exercised or were physically active. The answer options were “Yes” or “No”. Next, participants were asked to identify the physical activity or sport in which they exercised. They were provided with several options (ball games, power sports, high mass sports, aesthetic sports, weight-class sports, gravitational technical sports, endurance sports, recreational sports in gyms, individual indoor and outdoor recreational sports, group fitness activities without music and choreography-based group fitness activities with music and choreography, exercise at home). Based on the answers, four groups of sport participation were formed: organized team sports, organized individual sports, recreational sports and exercising at home. 

Body mass index (BMI) was calculated as body weight in kilograms (kg) divided by height in metres squared (m^2^) using self-reported information on body weight and height. In accordance with the World Health Organization’s suggested classification, students’ BMIs were classified into categories: underweight (<18.5 kg/m^2^), normal weight (18.5–24.9 kg/m^2^), overweight (25.0–29.9 kg/m^2^) and obese (≥30.0 kg/m^2^) [54]. The BMIs ranged from 14.9 to 38.0 (mean = 23.1, SD = 3.3) kg/m². The majority of the participants were of normal body weight (71.8%); 4.6% were underweight, 20.4% were overweight, and 3.2% were obese.

### 2.4. Statistical Analysis

Preliminary analyses and correlation analyses, as well as testing the variables’ distribution normality and the internal consistency of the scales, were conducted using SPSS v.29 (IBM Corp., Armonk, NY, USA). A Cronbach’s α over 0.65 was considered adequate [55], while it should generally be noted that Cronbach’s α values are sensitive to the number of items included in the scale [56]. After confirming the distribution normality of all the continuous variables, the Pearson correlation coefficient was used to test the associations between the SMS-PA-2 subscales, the total score and other study measures employed. Correlations between 0.1 and 0.3 were considered small, while those above 0.3 and below 0.5 were considered moderate, with a significance level of <0.05 [57]. Next, the means of the SMS-PA-2 subscales and the total score were compared in different sports groups and effect sizes were assessed via eta squared. A Tukey post hoc test was used for multiple pairwise comparisons between groups. The effect sizes, represented by eta squared, were calculated. An effect size above 0.01 and below 0.06 was considered small, above 0.06 and below 0.12 was seen as moderate, and ≥0.12 was considered large [57].

Finally, EFA and CFA with multi-group analysis for invariance testing were run using Mplus v8.7 (Muthén & Muthén, Los Angeles, CA, USA). The cut-off values for each model fit index were used as recommended by Hu and Bentler: RMSEA ≤ 0.06 for good fit and ≤0.08 for acceptable fit; SRMR ≤ 0.08 for good fit and ≤0.12 for acceptable fit; CFI ≥ 0.95 for good fit and ≥0.90 for acceptable fit [58].

## 3. Results

Descriptive characteristics of the SMS-PA-2 total score and its subscales are provided in Table 1. Mean scores of the subscales ranged from 2.85 (Accepting Mind) to 3.77 (Monitoring Body). All the variables were normally distributed, demonstrating small values of skewness and kurtosis.

Factor loadings from the exploratory (EFA) and confirmatory (CFA) factor analyses are presented in Table 2. First, to indicate the optimal number of factors of the Lithuanian translation of the SMS-PA-2, the EFA was run. The Kaiser–Meyer–Olkin test (KMO) resulted in a measure of sampling adequacy of 0.86, and Bartlett’s test of sphericity (χ^2^ = 4105.4, df = 171, *p* < 0.001) indicated the appropriateness of proceeding with the EFA. We used the varimax method to obtain orthogonal factors. Using this method, a four-factor solution with eigenvalues ≥ 1 was revealed with factor loading values of 0.46–0.81. None of the items were overlapping, and all of them were assigned to the original factors recommended by the scale authors. The four-factor model accounted for 58.8% of the total variance (the % of the variance explained by each factor is provided in Table 2). The next column of Table 2 represents standardized factor loadings from the CFA with the adequate values (0.56–0.77), except for item no. 15 (0.20).

The four-factor structure identified via EFA was next evaluated through CFA and indicated acceptable model fit to the data (Table 3). After removing item no. 15 with the low factor loading, a slightly improved model fit was observed. Also, acceptable model fit indices in sex groups were confirmed. Invariance analyses across sex groups revealed an acceptable fit of the configural, metric and scalar model. However, in the multi-group analysis, according to the χ^2^ test difference, metric invariance (metric against configural model χ^2^ (df = 14) = 32.54, *p* = 0.0034) and scalar invariance (scalar against metric χ^2^ (df = 14) = 24.99, *p* = 0.0347) were not confirmed. Metric invariance supports similar factor loadings across men and women groups, but these thresholds were non-invariant.

Table 4 exhibits Pearson correlation coefficients between SMS-PA-2 subscales and study measures. As expected, small-to-medium correlations were observed between SMS-PA-2 subscales and the total score and most study measures used in the analysis. The correlations between the SMS-PA-2 subscales and the BREQ-2 subscales representing amotivation and controlled exercise regulation were negative, while identified and intrinsic exercise regulation were positively correlated with the SMS-PA-2 total score and its subscales. Stronger correlations were observed between study measures and SMS-PA-2 subscales representing monitoring and accepting the body during the last exercise session. Finally, between monitoring and accepting the body and the total SMS-PA-2 score and disordered eating behaviour demonstrated by the EDE-Q-6 total score, small negative correlations were found. 

In Table 5, a comparison of the mean scores of the SMS-PA-2 total score and its subscales across groups of different sports is provided. Significant differences were found when comparing Monitoring Body, Accepting Body and the total SMS-PA-2 scores across sports groups. Home exercisers and persons engaged in team sports demonstrated lower levels of mindfulness during the last session of physical activity than those engaged in organized individual and recreational sports. According to the values of the eta squared of the differences obtained, small-to-medium effect sizes were observed (0.03–0.08). Moreover, we compared mean values of the SMS-PA-2 total score and its subscales across body mass index groups; no significant differences were found between BMI groups.

## 4. Discussion

The present study examined the psychometric properties of the Lithuanian translation of the SMS-PA-2 in a sample of physically active student-aged women and men. We expected that the Lithuanian version of the SMS-PA-2 would replicate the original four-subscale structure [1]. The psychometric analysis provided support for the Lithuanian version of the SMS-PA-2 with replication of the original factor structure supporting the use of either one general factor or four subscales representing monitoring of mind and body and accepting mind and body in physical activity experience. The SMS-PA-2 is based on MAT, which separates monitoring and awareness of mind and body and non-judgmental acceptance of mind and body [1]. 

However, in the Lithuanian version of the instrument, the standardized factor loading of item 15 (“I did not react to my thoughts/emotions”/“Nereagavau į savo mintis/emocijas”) was not in an acceptable range, and slightly improved model fit was observed when item 15 was deleted. The internal consistency of the Accepting Mind subscale increased when item 15 was removed; thus, we recommend using only two items (13 and 14) for this subscale in future studies using the Lithuanian version of the instrument. The internal consistency of the general scale and subscales was in good and acceptable ranges. Thus, these results provide promising evidence for future use of the SMS-PA-2 in MAT-based studies with young adult populations. Having a reliable and national language valid instrument would let us assess the correlates of mindfulness in different groups, contexts and levels of competence and observe the changes in state mindfulness as the outcome of various interventions.

Our study did not confirm complete measurement invariance between sexes. Acceptable configural, metric and scalar models demonstrated acceptable model fit indices, but the chi-square test comparing metric against configural and scalar against metric models demonstrated significant differences. It is important to continue testing SMS-PA-2 invariance between sexes in future studies to understand whether mindfulness differences in women and men reflect true attitudinal differences between sexes or psychometric differences related to item responses. In testing the validity of the original SMS-PA-2, the authors of the measure did not assess the measurement invariance across sexes [1]; therefore, our discussion is limited on this issue. Nevertheless, they recommend that future studies should assess it in the groups of sex, and also in the groups of physical activity type, length and experience [1]. 

Next, in the present study, we aimed to evaluate the construct validity of the instrument testing correlations between the SMS-PA-2 and trait awareness, physical activity, autonomous motivation in physical activity, positive body image, disordered eating attitudes and behaviours and self-esteem. In line with our hypothesis, we observed that the general score of the scale and four subscales of the SMS-PA-2 positively correlated with the Awareness subscale of the PHLMS. Also, positive small-to-medium correlations between physical activity and physical activity habits were observed, suggesting that physical activity and higher physical activity habits are associated with higher mindfulness. Previous research showed that trait mindfulness is positively associated with physical activity [15], and positive associations between mindfulness and physical activity habits were observed [59,60]. Therefore, the results of the present study make an important contribution to the field, suggesting that state mindfulness in physical activity is associated not only with greater physical activity but also habitual exercise behaviour. This might be explained by increases in trait mindfulness in physical activity that might be developed from a higher state of mindfulness in the long term. However, in the present study, we did not measure trait mindfulness in physical activity, and future studies are recommended to do this [15]. 

In the present study, state mindfulness was positively associated with more autonomous behavioural regulations of motivation and negatively related to controlled forms of motivation such as external regulation. Intrinsic motivation means that individuals engage in physical activity because the activity brings internal satisfaction, joy and enjoyment. In identified regulation, an individual associates physical activity with personal values and interests. Both regulations help to satisfy BPNs. Previous studies suggested that mindfulness might help to satisfy BPNs, especially feelings of autonomy and competence through non-judgmental awareness and acceptance of mind and body, increased satisfaction with physical activity, self-control and self-regulation [1,21,61]. 

Stronger correlations were observed between Monitoring Body and Accepting Body and physical-activity-related variables than between Monitoring and Accepting Mind. These results suggest that physical activity might be related to mindfulness not only via increases in physical activity motivation, self-regulation, self-control and physical activity satisfaction, but also through increases in positive embodiment [62]. Embodying means being in, or being associated with, the body or attunement to inner states, in contrast to body appearance measures that assess one’s externalized evaluation of body weight, body appearance and/or physical fitness [63]. Scholars suggest that participation in mindful physical activity might increase positive embodiment outside the mindful physical activity context, and these stable embodying experiences may then promote mindful self-care that might be related to maintaining physical activity [64]. However, in the present study, we did not assess whether physical activity was mindful or not; nevertheless, our results support the idea that mindfulness might be related to physical activity through increased monitoring and acceptance of the body. Future studies should test these assumptions.

In the present study, the SMS-PA-2 positively correlated with body appreciation and body functionality. Previous studies showed that trait and state mindfulness are associated with body appreciation [1,14]. The findings of our study make an important novel contribution, suggesting that state mindfulness in physical activity is associated with higher trait body functionality. This association might be explained by the process of possible positive embodiment in physical activity that is explained above. Individuals that are high in monitoring and awareness of mind and body might shift their attention from body appearance towards internal feelings and the functions of the body when performing exercise activity. Previous studies showed that state mindfulness is negatively associated with state body surveillance, which might be considered the opposite of body functionality appreciation [1]. Physical activity might positively affect trait body functionality through increased state mindfulness over prolonged periods of time. However, future studies of designs different than cross-sectional ones should test this assumption. Another explanation of the associations between higher state mindfulness in physical activity and body appreciation is increased interoceptive awareness. A recent study reported positive associations between interoceptive awareness and positive body image in British adults [65]. Future studies should test these associations in exercisers. 

Further, the SMS-PA-2 scale was negatively related to disordered eating attitudes and behaviours. Mindfulness of mind and body subscales were not associated with EDE-Q-6, yet mindfulness of body and mind were weakly negatively associated with it. These findings are in line with previous studies suggesting that mindfulness is inversely related to binge eating and emotional eating [30]. The associations between state mindfulness in physical activity and disordered eating might be related through the increases in positive body image, which is negatively associated with disordered eating [66]. In other words, our results suggest that high levels of monitoring and awareness of body sensations in physical activity might lead individuals to higher general individual sensitivity to body needs and sensations and higher acceptance of the body and lower body dissatisfaction, behavioural automaticity and uncontrolled eating. 

The final aim of the present study was to compare SMS-PA-2 scores in groups of different types of sport involvement. The results of the present study show that individuals who exercised at home and participants of organized team sports demonstrated lower levels of state mindfulness than those engaged in organized individual or recreational sports. These results partially support the results of a previous study in which higher mindfulness was observed in organized-sports participants than in those who did not participate in any sports [24]. Exercising at home might be interrupted by various interferences that distract attention from feelings of the body to the external environment. The theory of cue competition hypothesis states that competition between internal and external stimuli exists and paying attention to external cues will reduce the attentional resources available for internal cues [67,68]. 

Another interesting finding was that mindfulness was lower in participants of organized team sports than in participants of organized individual sports. This is an important new contribution to the literature. The nature of team sports is different from that of individual sports since typical sports games activities are highly oriented towards teams winning, and individual mindfulness might be less promoted in this type of exercise. Future studies should test both our findings and psychological mechanisms related to the state mindfulness differences between team and individual sports. 

Several important limitations should be considered when discussing the results of this study. First, the SMS-PA-2 scale should be tested immediately after undertaking a physical activity. The original version of the scale was first tested using this approach. In our study, we asked participants to provide answers based on recalling the last exercise or physical activity session they had participated in. Half of the sample had exercised on the same or previous day, more than a third of the sample within one week and more than 10% over one week earlier. It might be that the longer the time between the exercise workout and testing, the greater the chance of discrepancy and error. Thus, future studies should be implemented testing the SMS-PA-2 immediately after a physical activity session. Another limitation is that the sample was homogeneous and represented young healthy physically active students. Future studies should involve more diverse populations. The directions of association in the present study are not clear since this study is cross-sectional. Future studies with different designs other than cross-sectional ones should test our findings. Finally, the sample size was too low to divide it for EFA and CFA tests separately. Thus, in future studies, it is essential to overcome this limitation.

## 5. Conclusions

The Lithuanian version of the SMS-PA-2 is a reliable and valid instrument for measuring state mindfulness in physical activity, and it is recommended for further use in Lithuanian-speaking samples of young adults. This instrument is recommended for researchers aiming to investigate the role of state mindfulness in the practice of physical activity and sport, to explore the changes in motivation and positive body image in physical activity and to test the effectiveness of mindfulness-based physical activity interventions focused on the development of various health-related outcomes.

## Figures and Tables

**Table 1 behavsci-13-00820-t001:** Descriptive characteristics of the Lithuanian translation of the State Mindfulness Scale for Physical Activity-2 (SMS-PA-2), *n* = 539.

Subscales	Mean	SD	Median	Skewness	Kurtosis	Min.	Max.
MM	3.21	0.84	3.17	−0.36	0.08	1.0	5.0
MB	3.77	0.80	3.83	−0.48	0.05	1.0	5.0
AM	2.85	0.87	3.00	0.32	0.01	1.0	5.0
AB	3.47	0.79	3.50	−0.15	−0.24	1.3	5.0
SMS-PA-2 total score	3.38	0.59	3.37	−0.15	0.41	1.5	5.0

MM—Monitoring Mind, MB—Monitoring Body, AM—Accepting Mind, AB—Accepting Body, SD—standard deviation, min.—minimum, max.—maximum scored value.

**Table 2 behavsci-13-00820-t002:** Factor loadings from the exploratory (*n* = 269) and confirmatory (*n* = 270) factor analyses in the random split-half student samples of the Lithuanian translation of the State Mindfulness Scale for Physical Activity-2.

Statements	Factor Loadings from the EFA				Standardized factor Loadings from the CFA			
	MM	MB	AM	AB	MM	MB	AM	AB
1. I was aware of different emotions that arose in me	0.71				0.69			
2. I noticed pleasant and unpleasant emotions	0.69				0.64			
3. I noticed pleasant and unpleasant thoughts	0.79				0.56			
4. I noticed emotions come and go	0.77				0.67			
5. I noticed thoughts come and go	0.74				0.65			
6. It was interesting to see the patterns of my thinking	0.68				0.63			
7. I focused on the movement of my body		0.58				0.65		
8. I felt present in my body		0.78				0.66		
9. I listened to what my body was telling me		0.76				0.73		
10. I was aware of how my body felt		0.79				0.75		
11. I noticed the sensations in my body		0.78				0.78		
12. I was in tune with how hard my muscles were working		0.59				0.67		
13. I let my thoughts/emotions just be without fixating on them			0.46				0.75	
14. I accepted my thoughts/emotions without judging them			0.79				0.62	
15. I did not react to my thoughts/emotions			0.81				0.20	
16. I was okay with experiencing the physical sensations in my body				0.66				0.62
17. I acknowledged how my body felt without trying to change it				0.80				0.77
18. I accepted how my body felt even if it was unpleasant				0.67				0.66
19. I was okay with how my body felt, even if it did not meet expectations				0.67				0.62
% of the variance explained	18.2	19.6	8.2	12.8				

Factor loadings < 0.40 for the EFA are not shown; EFA—exploratory factor analysis, CFA—confirmatory factor analysis; MM—Monitoring Mind, MB—Monitoring Body, AM—Accepting Mind, AB—Accepting Body.

**Table 3 behavsci-13-00820-t003:** The State Mindfulness Scale for Physical Activity-2 (SMS-PA-2) confirmatory analysis model fit indices and measurement invariance across sex groups.

Models	χ^2^	df	CFI	RMSEA	90% CI	SRMR
Original 4-factor model (19 items)	473.1	143	0.92	0.065	0.059–0.072	0.060
4-factor model without statement 15 (18 items)	338.8	126	0.95	0.056	0.049–0.063	0.048
4-factor model in men only (18 items)	230.4	126	0.94	0.055	0.044–0.066	0.053
4-factor model in women only (18 items)	281.0	126	0.93	0.068	0.057–0.078	0.059
Configural model	511.3	252	0.94	0.062	0.054–0.069	0.056
Metric model	543.9	266	0.93	0.062	0.055–0.070	0.066
Scalar model	568.8	280	0.93	0.062	0.055–0.069	0.066

CFI—comparative fit index, RMSEA—root mean square error of approximation, SRMR—standardized root mean square residual, df—degree of freedom, CI—confidence interval.

**Table 4 behavsci-13-00820-t004:** Correlations between study measures and the Lithuanian translation of the State Mindfulness Scale for Physical Activity-2 (SMS-PA-2), *n* = 539.

Study Measures	MM	MB	AM	AB	SMS-PA-2 Total Score
Godin LTEQ	0.09 *	0.18 **	0.06	0.18 **	0.18 **
PA habits	0.18 **	0.28 **	0.21 **	0.27 **	0.31 **
BREQ-2: Amotivation	–0.04	–0.20 **	–0.05	–0.20 **	–0.17 **
BREQ-2: External	–0.03	–0.21 **	–0.11 *	–0.22 **	–0.19 **
BREQ-2: Introjected	0.11 *	0.06	–0.01	0.04	0.08
BREQ-2: Identified	0.15 *	0.34 **	0.14 **	0.25 **	0.29 **
BREQ-2: Intrinsic	0.15 *	0.38 **	0.16 **	0.32 **	0.35 **
FAS	0.17 **	0.45 **	0.21 **	0.40 **	0.42 **
BAS-2	0.16 **	0.43 **	0.21 **	0.43 **	0.42 **
Rosenberg SES	0.10 *	0.35 **	0.17 **	0.36 **	0.32 **
PHLMS Awareness	0.28 **	0.43 **	0.22 **	0.36 **	0.45 **
EDE-Q-6	0.02	–0.14 **	–0.04	–0.23 **	–0.12 **

* *p* < 0.05, ** *p* < 0.01; MM—Monitoring Mind, MB—Monitoring Body, AM—Accepting Mind, AB—Accepting Body, LTEQ—Leisure-Time Exercise Questionnaire, PA—physical activity, BREQ-2—Behavioural Regulation in Exercise Questionnaire-2, FAS—Functionality Appreciation Scale, BAS-2—Body Appreciation Scale-2, SES—Self-Esteem Scale, PHLMS—Philadelphia Mindfulness Scale, EDE-Q-6—Eating Disorder Examination Questionnaire.

**Table 5 behavsci-13-00820-t005:** Comparison of the State Mindfulness Scale for Physical Activity-2 (SMS-PA-2) mean scores (mean ± SD) across different sports groups (*n* = 538).

Subscales	MM	MB	AM	AB	SMS-PA-2 Total Score
Organized team sports, *n* = 117	3.15 ± 0.80	3.44 ± 0.78	2.86 ± 0.98	3.33 ± 0.77	3.25 ± 0.60
Organized individual sports, *n* = 149	3.34 ± 0.93	3.99 ± 0.77 ^a^	3.08 ± 1.05	3.63 ± 0.75 ^a^	3.59 ± 0.63 ^a^
Recreational sports, *n* = 181	3.21 ± 0.79	3.89 ± 0.75 ^a^	3.07 ± 1.00	3.54 ± 0.77 ^a^	3.49 ± 0.58 ^a^
Home exercise, *n* = 91	3.10 ± 0.83	3.59 ± 0.82 ^b,c^	2.92 ± 0.89	3.27 ± 0.87 ^b,c^	3.28 ± 0.64 ^b,c^
*p*	0.139	< 0.001	0.327	0.001	< 0.001
Eta squared	-	0.08	-	0.03	0.05

MM—Monitoring Mind, MB—Monitoring Body, AM—Accepting Mind, AB—Accepting Body; ^a^ *p* < 0.05 as compared to organized sports group, ^b^ *p* < 0.05 as compared to individual sports group, ^c^ *p* < 0.05 as compared to recreational sports group.

## Data Availability

The data set generated and analysed during the current study is available from the corresponding author, Migle Baceviciene.

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
