# Peer review of "Mindfulness, Physical Activity and Sports Participation: Validity Evidence of the Lithuanian Translation of the State Mindfulness Scale for Physical Activity 2 in Physically Active Young Adults"

_behavsci, 2023, doi:10.3390/bs13100820_

Round 1

Reviewer 1 Report

The submission reports on a contemporary topic and provides relevant background theory helping to give conceptual backbone to the measure being developed. Very clear reporting within the Methods – e.g., explicit inclusion criteria, questionnaire completion rates. Findings and the Discussion are also presented clearly. The following comments are intended to help the authors further the write up of this research.

1.      I believe that there is rationale for the present study, but this could be further refined. For example, lines 168-170, please can you expand to clarity why more studies are needed.

2.      Generally, clear aims are presented although the justification for the final aim involving a comparison of SMS-PA-2 scores across types of sport involvement (organized team sports, organized individual sports, recreational sports and home exercise) needs some bolstering.

3.      Given that the study recruited from Lithuania I doubt that only the SMS-PA-2 translated, wouldn’t all the measures used in the investigation need to be Lithuanian including the Philadelphia Mindfulness Scale?

4.      Use of data from the BREQ-2 to create a composite score via the relative autonomy index (RAI) formula is suboptimal and somewhat dated. I can understand why the researchers might have opted for this approach, but I think that this needs to be either some discussion as a limitation (as autonomous and controlling forms of motivation might be more appropriate) or re-analysis of these mechanism related data.

5.      With regard to the order of the analytic strategy, wouldn’t it be more logical to address factorial (internal) validity before assessing correlations (external validity) with other scales?

6.      What was the conceptual rationale for the removal of item 15 from the CFA?

7.      Sorry if I missed this but I think that rationale for the multi-group invariance analysis is absent.

8.      Would it be more appropriate to term the gender invariance analysis as a sex invariance analysis? Also, can the findings from this analysis be contextualised with the existing literature – has this been examined before, are the current findings a surprise?

Author Response

Dear Reviewer, 

Thank you for your time reviewing our paper and for your valuable comments. Please see our responses below. All the changes made in the text are highlighted in a blue font.

Comments and Suggestions for Authors

The submission reports on a contemporary topic and provides relevant background theory helping to give conceptual backbone to the measure being developed. Very clear reporting within the Methods – e.g., explicit inclusion criteria, questionnaire completion rates. Findings and the Discussion are also presented clearly. The following comments are intended to help the authors further the write up of this research.

  1.     I believe that there is rationale for the present study, but this could be further refined. For example, lines 168-170, please can you expand to clarity why more studies are needed.

Exploring the associations between state mindfulness, physical activity and body image is a very new research area, there is a lack of research in this area and the majority of them were implemented using Western European samples. Therefore, it is important to continue this type of research. This text was included in the recommended place.

  1. Generally, clear aims are presented although the justification for the final aim involving a comparison of SMS-PA-2 scores across types of sport involvement (organized team sports, organized individual sports, recreational sports and home exercise) needs some bolstering.

We included an argument that helped to justify why we decided to compare the sport involvement groups:

Since there is a lack of research assessing mindfulness in various physical activities and sports [18], the final aim of the present study was to compare SMS-PA-2 scores in the groups of different types of sport involvement (organized team sports, organized individual sports, recreational sports and home exercise).  

  1. Given that the study recruited from Lithuania I doubt that only the SMS-PA-2 translated, wouldn’t all the measures used in the investigation need to be Lithuanian including the Philadelphia Mindfulness Scale?

All measures used were Lithuanian translations that were previously validated in Lithuanian languages. We included this information in the Methods section.

  1. Use of data from the BREQ-2 to create a composite score via the relative autonomy index (RAI) formula is suboptimal and somewhat dated. I can understand why the researchers might have opted for this approach, but I think that this needs to be either some discussion as a limitation (as autonomous and controlling forms of motivation might be more appropriate) or re-analysis of these mechanism related data.

Thank you for this important comment. We re-analyzed the data and included new information using the subscales of the BREQ-2.

  1. With regard to the order of the analytic strategy, wouldn’t it be more logical to address factorial (internal) validity before assessing correlations (external validity) with other scales?

Thank you for this comment. Factorial (internal) validity was presented before assessing correlations (external validity) with other scales.

  1. What was the conceptual rationale for the removal of item 15 from the CFA?

Item 15 demonstrated very low factor loading in the CFA model. Moreover, after removing it, improved model fit indices were observed.

  1. Sorry if I missed this but I think that rationale for the multi-group invariance analysis is absent.

Thank you for this important comment. Just because a model has been demonstrated to have a good fit does not mean that the response to individual items can be explained by the same latent factors. As recommended by V. Swami and D. Baron (Body Image, 2019), multi-group CFA should be performed and examined consecutively at the configural, metric and scalar levels for all translated instruments.

  1. Would it be more appropriate to term the gender invariance analysis as a sex invariance analysis? Also, can the findings from this analysis be contextualised with the existing literature – has this been examined before, are the current findings a surprise?

The term „gender invariance“ was replaced by the „sex invariance“.

The invariance between sexes was not assessed in the original version of the SMS-PA-2, therefore, the discussion on this topic is limited. However, we included this information in the Discussion.

Reviewer 2 Report

Dear authors,

I congratulate you on the work you have presented on an important topic with an impact on the field.

However, in order for the version to be accepted, I suggest some minor adjustments.

1. Consider the recommendation in the abstract. It suggests that the Lithuanian version can be used in different contexts. I suggest inserting "in Lithuanian-speaking countries/populations of young adults.

2. Also, in the abstract, I recommend another consideration, to include that the scale be used immediately after a physical activity session. In addition to the technical aspect, I think the operational indication is important.

3. Apart from the authors' care in introducing the central questions of the study, I think the introduction is too long and tiresome. I would make the introduction shorter, making the methods and results more attractive to read. The elements of the introduction are essential to strengthen the discussion.

Author Response

Dear Reviewer, 

Thank you for your time reviewing our paper and for your comments. Please see our responses below. All the changes made in the text are highlighted in a blue font.

Dear authors,

I congratulate you on the work you have presented on an important topic with an impact on the field. However, in order for the version to be accepted, I suggest some minor adjustments.

  1. Consider the recommendation in the abstract. It suggests that the Lithuanian version can be used in different contexts. I suggest inserting "in Lithuanian-speaking countries/populations of young adults.

Thank you, we revised the text accordingly in the abstract.

  1. Also, in the abstract, I recommend another consideration, to include that the scale be used immediately after a physical activity session. In addition to the technical aspect, I think the operational indication is important.

Thank you, we revised the text accordingly in the abstract.

  1. Apart from the authors' care in introducing the central questions of the study, I think the introduction is too long and tiresome. I would make the introduction shorter, making the methods and results more attractive to read. The elements of the introduction are essential to strengthen the discussion.

Thank you for this comment. Since the research of state mindfulness is quite new in physical activity and body image contexts, we decided to keep all the text without shortening it to provide clarity of the rationale of the study. Other reviewers did not recommend shortening the Introduction. However, we double-checked the Introduction and minimally revised some parts of it.